# LogiGAN: Learning Logical Reasoning via Adversarial Pre-training

**Xinyu Pi**[1]*, **Wanjun Zhong**[2]*, **Yan Gao**[3], **Nan Duan**[3], **Jian-Guang Lou**[3]

[1]University of Illinois Urbana-Champaign, Urbana, USA
[2]Sun Yat-Sen University   [3]Microsoft Research Asia
xinyupi2@illinois.edu, zhongwj25@mail2.sysu.edu.cn
{yan.gao, jlou, nanduan}@microsoft.com

## Abstract

We present LogiGAN, an unsupervised adversarial pre-training framework for improving logical reasoning abilities of language models. Upon automatic identification of logical reasoning phenomena in massive text corpus via detection heuristics, we train language models to predict the masked-out logical statements. Inspired by the facilitation effect of reflective thinking in human learning, we analogically simulate the learning-thinking process with an adversarial Generator-Verifier architecture to assist logic learning. LogiGAN implements a novel sequential GAN approach that *(a)* circumvents the non-differentiable challenge of the sequential GAN by leveraging the Generator as a sentence-level generative likelihood scorer with a learning objective of reaching scoring consensus with the Verifier; *(b)* is computationally feasible for large-scale pre-training with longer target length. Both base and large size language models pre-trained with LogiGAN demonstrate obvious performance improvement on 12 datasets requiring general reasoning abilities, revealing the fundamental role of logic in broad reasoning, as well as the effectiveness of LogiGAN. Ablation studies on LogiGAN components reveal the relative orthogonality between linguistic and logic abilities and suggest that reflective thinking's facilitation effect might also generalize to machine learning [2].

## 1   Introduction

*"Learning without thinking is labor lost; thinking without learning is perilous."*   – *Confucius*

Pre-trained Language Models (PLMs) (Devlin et al., 2018; Brown et al., 2020; Raffel et al., 2020) are approaching human-level performance in numerous tasks requiring basic linguistic abilities (Rajpurkar et al., 2016; Wang et al., 2018), setting off a huge wave of interest in Natural Language Processing (NLP). Despite the emerging fervor, researchers soon realized that PLMs are relatively incompetent in their **reasoning** abilities, which seems to be an insurmountable bottleneck for PLMs with even better linguistic abilities (Kassner & Schütze, 2019; Helwe et al., 2021). Following this, researchers delve into reasoning from multitudinous aspects, striving to improve PLMs' reasoning abilities.

From our perspective, reasoning (in natural language) is essentially an inferential process where an unstated statement is drawn based on several presented statements, and **Logic** is the systemic set of principles that provides reasoning with correctness and consistency assurance (Hurley, 1982). Regardless of the variability of contents, logical reasoning generally incorporates two invariant forms: drawing conclusions based on some premises (aka. deduction & induction, (Reichertz, 2013)), or

---

*indicates equal contribution. Work done during internship at Microsoft Research Asia.

[2]The code is released in `https://github.com/microsoft/ContextualSP/tree/master/logigan`

hypothesizing premises to explain some conclusions (aka. abduction (Douven, 2021)). Most existing tasks requiring general reasoning ability, such as natural language inference (Nie et al., 2019) and complex machine reading comprehension (Lai et al., 2017), can be readily interpreted by this criterion. Other tasks requiring specialized reasoning skills can be considered either as (i) providing sufficient premises but requiring specific ways of premise extraction to draw conclusions, such as multi-hop (Yang et al., 2018b) or hybrid (Chen et al., 2020) reasoning; or (ii) requires external knowledge, such as commonsense (Sap et al., 2020) or numerical (Dua et al., 2019) knowledge, as premises to draw conclusions, hence could also be interpreted by the two forms of logical reasoning. Following this analysis on the relation between logic and reasoning, *Logic ability* will be an essential foundation for a broad scope of reasoning, and should be prioritized in improving PLMs' reasoning abilities [3].

Conventional pre-training via *randomized* **M**asked **L**anguage **M**odeling (MLM) and auxiliary tasks are generally developed upon Firth (1957)'s distributional hypothesis of semantics – "a word is characterized by the company it keeps." Under this paradigm, models efficiently learn to capture grammatical structures and contextualized semantics. However, since logical consistency is beyond correctness on a linguistic level, it is less obvious how MLM could help with logical reasoning abilities. Do models harvest logic ability for free from MLM? Or is that something that needs further learning beyond language acquisition? Motivated by these questions, we propose an **unsupervised pre-training method aiming at enhancing the logical reasoning ability of PLMs**: we automatically identify occurrences of logical reasoning phenomena in large corpus via detection heuristics, and then require PLMs to predict the masked-out logical statements made in the original context (Section 3). For example, in the case "Bob recently made up his mind to lose weight. *Therefore*, `[MASK]`", the prediction goal is the masked original statement "he decides to go on a diet".

However, statements different from the original statement could also be logically consistent with respect to a given context. For example, "he decides to exercise from today on" is also a reasonable inference in the case above. Since Generators trained merely from recovering original statements are not encouraged to explore the possibilities of other reasonable statements, their overall learning effectiveness of logic could potentially be degraded. Therefore, it makes sense to provide additional feedback based on the degree of logical consistency between statements predicted beforehand and the original context – we realize this much resembles humans' reflective thinking process. Inspired by research from cognitive psychology (Moon, 2013; Boud et al., 2013; Di Stefano et al., 2016) advocating for the vital role of reflective thinking in improving the experiential efficiency of human learning, we hypothesize that machines might also benefit from reflective thinking in their learning processes. Following this hypothesis, we analogically simulate humans' learning-thinking process with a Generator-Verifier architecture, and propose **LogiGAN**, a novel adversarial training approach tailored for sequential GAN training to further facilitate the learning of logical reasoning.

In LogiGAN's design, the Generator learns not only to recover the original masked statements, but also to score candidate statements (based on their generative likelihood) in a manner that could reach scoring consensus with the Verifier, who learns to make judgments on the logical consistency between premises and conclusions. The more logically consistent the Verifier thinks of a statement w.r.t. the input context, the higher generative likelihood score is expected to be assigned by the Generator. To encourage the exploration of broader possibilities of reasonable statements other than the original one, we also apply a diversified sampling strategy for candidate statement generation. Both Generator and Verifier scoring processes are continuous throughout the adversarial training, thereby circumventing the non-differentiable barrier in sequential GAN posed by the discrete beam-search. Moreover, LogiGAN does not involve token-wise Monte Carlo Search for policy gradient estimation, and scoring processes of Generator and Verifier are decoupled, so that parallel score computation is possible. This makes large-scale pre-training with longer target length computationally feasible.

To test the effectiveness of LogiGAN, we extensively experiment on **12** datasets requiring general reasoning ability. The apparent performance improvements of *both base and large size PLMs* across all tasks reveal models' harvest of logic ability, shoring up the fundamental role of logic in general reasoning. We also carry out ablation studies to understand the functionality of LogiGAN components, the results of which shed light on the relative orthogonality between linguistic and logic ability and suggest that the facilitation effect of reflective thinking is also generalizable to machine learning.

---

[3]We expand this analysis in-depth in App. A, and refer intrigued readers there.

## 2 Logic Pre-training

In this work, we primarily focus on improving PLMs' ability of *informal logic* (Groarke, 2021). We include the three most classical types of logical reasoning: **deductive, inductive, and abductive reasoning** conducted in the form of natural language (Reichertz, 2004; Kennedy & Thornberg, 2018; Reichertz, 2007; Douven, 2021) in our consideration. Note that our coverage is broader than the informal logic strictly defined in the philosophy community (Munson, 1976) that primarily focuses on analyzing the soundness and cogency of the application of the aforementioned reasoning in real-life arguments. The other half of logic investigation – the normative study of formal logic (typically conducted in a symbolic form), such as truth-function logic (Buvac & Mason, 1993), modal logic (Priest, 2008), and fuzzy logic (Dote, 1995), is beyond the scope of this paper.

**Logic Indicators as Detection Heuristics.** To set up an unsupervised pre-training aiming at improving models' logic ability, the very first step will be to identify logical reasoning phenomena from a vast-scale unstructured text corpus. While invocations of logic are not explicitly stated in most cases, writers' usage of *logic indicators* usually marks their logical reasoning processes with high precision (Hurley, 1982), thereby serving as an ideal heuristic device for our detection purpose. We consider two standard types of logic indicators: **(i)** *conclusion indicator* such as "Therefore", "We may infer that", which denotes drawing conclusion deductively or inductively from given premises; And **(ii)** *premise indicator* such as "Due to", "The reason that", which denotes abductively hypothesizing premises that explain or provide evidence to some stated conclusions.

**Corpus Construction.** For a text corpus, we detect every occurrence of pre-defined logic indicators (listed in App. C), and mask out (i.e., replace with `[MASK]`) the entire **statement** subsequent to the indicator (each training example will have exactly one masked-out statement). Then models' task will be to perform language modeling and predict the masked statement. We emphasize that **statements** are declarative sentences or declarative clauses, owning complete subject and predicate structures, and are capable of being factually true or false. To supply sufficient context information for these predictions, we keep $x$ complete sentences previous to the `[MASK]`, as well as $y$ sentences after the `[MASK]`, where $x$ and $y$ can be sampled from a geometric distribution with pre-defined hyper-parameters. Fig. 1 illustrates the input and output format, and we discuss details in Sec. 4.

**Masked Logical Statement Prediction.** In the simplest setting, the Generator learns to infill the masked statement via a *single-task* pre-training, which fulfills the training process of a typical masked language modeling task. The only difference is that models no longer predict *randomly masked tokens or spans* but instead *logic-targeted masked complete statements*. Models are trained to perform Max Likelihood Estimation (MLE) for masked statements under a typical teacher forcing loss. Practically, generative pre-trained language models such as T5 (Raffel et al., 2020) could take up the position of Generator $\mathcal{G}$. Given a single input context / output statement pair $(c, s)$, the teacher forcing loss can be mathematically expressed as [4]:

$$\mathcal{L}_{tf}(c, s) = -\frac{1}{T} \sum_{t=1}^{T} \log p_{\mathcal{G}_\theta} \left( w_t(s) \mid w_{1:t-1}(s); c \right) \qquad (1)$$

## 3 The Adversarial Training Framework

Since Generators trained merely from recovering masked original statements miss out opportunities of exploring other reasonable statements, LogiGAN implements an adversarial mechanism for providing Generators with extra signals based on logical consistency between pseudo-statements (sampled from Eq. 3) and context to encourage explorations. The adversarial framework has two major components: (i) a *Verifier* $\mathcal{V}$ that learns to judge logical consistency between statements and context; (ii) a *Generator* $\mathcal{G}$ that learns both to recover masked original statements, and scores pseudo-statements (based on their generative likelihood) in a manner that could reach scoring consensus with the Verifier – The more logically consistent the Verifier thinks of a statement w.r.t. the input context, the more likely the Generator is expected to generate the statement under the input context (i.e., assign higher generative likelihood score). The overall objective of LogiGAN can be expressed as the minimax objective:

$$J^{\mathcal{G}^* \cdot \mathcal{V}^*} = \min_\theta \max_\phi \ \mathbb{E}_{s^+ \sim p_{\text{true}}(.|c)} [\log \mathcal{V}_\phi(c, s^+)] + \mathbb{E}_{s^- \sim p_{\text{neg}}(.|\mathcal{G}_\theta, c, s^+)} [\log(1 - \mathcal{V}_\phi(c, s^-))]. \qquad (2)$$

---

[4] $w_t(.)$ denotes the $t^{th}$ token of a input string.

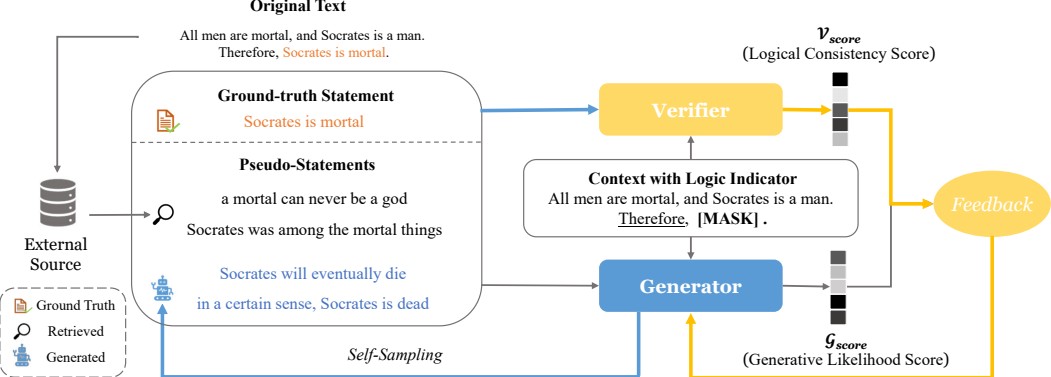

Figure 1: LogiGAN Overview. Generator targets to predict the masked-out logical statement and scores candidate statements, while Verifier justifies the logical correctness of statements. The blue path indicates the process where the Generator helps Verifier learning, while the yellow path denotes the process of giving Verifier feedback for Generator training.

where $\mathcal{G}_\theta$ and $\mathcal{V}_\phi$ denote the Generator and the Verifier with model parameters $\theta$ and $\phi$, respectively. $s^+/s^-$ represents ground-truth statements from original text / sampled pseudo-statement [5]. We discuss sampling details of pseudo-statements later in this section in Eq. 3.

Classical GAN settings (Goodfellow et al., 2014; Zhu et al., 2017) fall short in sequential generation because the gradient propagation from the Verifier to the Generator is blocked by a non-differentiable beam-search during text generation. Previous approaches such as (Yu et al., 2017) address this challenge by token-wise policy gradient estimation via Monte Carlo Search. However, since the sampling run-time grows exponentially with the length of the target sequence, their original implementations are not applicable to million-scale pre-training with relatively longer target length as in our scenario.

Different from them, LogiGAN omits the token-wise Monte Carlo Search for policy gradient estimation, and realizes a similar goal via measuring the similarity of scoring distributions between Verifier and Generator. The main procedures of LogiGAN can be summarized in four steps: *(a)* several candidate pseudo-statements are sampled on a sentence level; *(b)* the Verifier assigns the ***logical consistency scores*** $\mathcal{V}_{score}$ based on how logical consistent these candidates are w.r.t the original context; *(c)* the Generator assigns the sentence-level ***generative likelihood score*** $\mathcal{G}_{score}$ to each candidate to reflect how likely it will produce the pseudo-statement under the given context. *(d)* The similarity between Generator and Verifier score distributions is computed as a new signal to encourage the Generator to reach scoring consensus with the Verifier – i.e., the more logically consistent the Verifier thinks of the statement, the higher likelihood score the Generator is expected to assign. Since both scoring processes are continuous, the non-differentiable barrier is successfully bypassed. Meanwhile, this design does not involve sequential token-level sampling and decouples the Generator and Verifier scoring processes, thereby enabling parallel score computations. This makes large-scale pre-training with relatively longer target sequence length computationally feasible.

The overall framework overview is illustrated in Fig. 1, and the detailed training procedure is summarized in Algorithm 1. To diversify the candidate pseudo-statements, we sample pseudo-statements from two sources: (i) self-sampling via diversified beam-search from the Generator; or (ii) retrieving similar statements from the corpus, and the sampling process can be summarized as:

$$p_{\text{neg}}(. \mid \mathcal{G}_\theta, c, s^+) = \{s_\alpha \sim \mathcal{G}_\theta(. \mid c) \ \cup \ s_\beta \sim R(s^+)\}, \tag{3}$$

where $\mathcal{G}_\theta(. \mid c)$ denotes self-sampled statement $s_\alpha$ given context $c$ from Generator $\mathcal{G}_\theta$, and $R(s^+)$ denotes a retriever[6] that retrieves textually similar statements $s_\beta$ with ground-truth statement $s^+$ from the corpus. Note that this process is conducted separately for the corpus of Verifier and Generator.

---

[5]Note: in real practice, there is a **gap** between sampled *pseudo-statements* $s^-$ and *logically inconsistent* statements. We keep current symbolic denotations for simplicity only and discuss this issue in App. B.

[6]Any retriever is feasible and we adopt BM25 as the retrieving method here.

**Algorithm 1:** Adversarial Training Process

**Dependencies:** (1) A Pre-trained Generative Language Model as Generator $\mathcal{G}_0$
(2) A Pre-trained Discriminative Language Model as Verifier $\mathcal{V}_0$
(3) Generator Source Training Coprus $C_\mathcal{G}$ with size $M$
(4) Verifier Source Training Corpus $C_\mathcal{V}$ with size $N$
(5) Pre-defined Warmup epochs $E$, max iterations of GAN training $Q$
(6) Pre-defined training sample size $m$, $n$ for $\mathcal{V}$, $\mathcal{G}$ per iteration

1  Random partition $C_\mathcal{G}$ into $C_{\mathcal{G}_\alpha}$, $C_{\mathcal{G}_\beta}$ with size $M_\alpha$, $M_\beta$;
2  $\mathcal{G}_0 \leftarrow$ Warmup $\mathcal{G}_\alpha$ on $C_{\mathcal{G}0}$ for $E$ *epochs* with $\mathcal{L}_{tf}$;
3  **for** *i in 1:Q* **do**
4  $\quad \mathcal{G}_i \leftarrow \mathcal{G}_{i-1}$;
5  $\quad C_{\mathcal{V}i}, C_{\mathcal{G}_i} \leftarrow$ Sample $m$ examples from $C_\mathcal{V}$, and $n$ examples from $C_{\mathcal{G}_\beta}$, w/o replacement;
6  $\quad \widetilde{C_{\mathcal{V}i}}, \widetilde{C_{\mathcal{G}_i}} \leftarrow$ Sample pseudo-statements for $C_{\mathcal{V}i}, C_{\mathcal{G}_i}$ with $\mathcal{G}_i$ and BM25, as in Eq. 3;
7  $\quad \mathcal{V}_i \leftarrow$ Train $\mathcal{V}_{i-1}$ on $\widetilde{C_{\mathcal{V}i}}$ for 1 *epoch* with $\mathcal{L}_{ver}$, as in Eq. 4;  (Verifier Training)
8  $\quad$ **for** $\widetilde{c}$ *in batch (* $\widetilde{C_{\mathcal{G}_i}}$ *)* **do**
9  $\quad\quad \mathcal{V}_{score}, \mathcal{G}_{score} \leftarrow \mathcal{V}_i$, $\mathcal{G}_i$ do scoring on $\widetilde{c}$, as in Eq. 5 and 6;
10 $\quad\quad \mathcal{L}_{gen} \leftarrow \lambda_1 \mathcal{L}_{tf}(s^+ \text{ from } \widetilde{c}) + \lambda_2 D_{KL}(\mathcal{V}_{score} \,||\, \mathcal{G}_{score})$ , as in Eq. 7;
11 $\quad\quad \mathcal{G}_i \leftarrow$ Update $\mathcal{G}_i$ for 1 *step* with $\mathcal{L}_{gen}$;  (Generator Training)
12 $\quad$ **end**
13 **end**

## 3.1  Training of Verifier

The Verifier serves as a critic to judge whether a statement is logically consistent w.r.t. the context. Therefore, the training task of Verifier can be viewed as a binary classification problem. Pre-trained language models that could perform discriminative classification tasks such as BERT (Devlin et al., 2018), ALBERT (Lan et al., 2019), and RoBERTa (Liu et al., 2019), will be suitable for the role of Verifier. With $y = 1$ for both ground-truth and logically consistent pseudo-statements, and $y = 0$ for other pseudo-statements, the binary classification loss for a single pair of context/statement/label $(c, s, y)$ of Verifier can be mathematically expressed as (omitting average):

$$\mathcal{L}_{ver}(c, s, y) = -y \log \mathcal{V}_\phi(y \mid [c; s]) - (1 - y) \log(1 - \mathcal{V}_\phi(y \mid [c; s])), \tag{4}$$

## 3.2  Training of Generator

The Generator targets both to recover the original masked statements, and to score pseudo-statements based on their generative likelihood in a manner that could reach sentence-level scoring consensus with the Verifier. This corresponds to the two sources of learning signals received by the Generator: (i) the original generative objective with teacher forcing loss defined in Eq.1 as a signal; and (ii) the distribution similarity between sentence-level generative likelihood score assigned by Generator and logic consistency score assigned by Verifier. To achieve the goal of (ii), we first sample pseudo-statements $\{s_1^-, ..., s_n^-\}$ from $p_{neg}(. \mid \mathcal{G}_\theta, c, s^+)$. Then the Verifier assigns ***logical consistency score*** $\mathcal{V}_{score}$ based on how logically consistent the pseudo-statements are w.r.t. the context, expressed as:

$$\mathcal{V}_{score}(c; s_1^-, ..., s_n^-) = [\mathcal{V}_\phi(s_1^-, c); \ \mathcal{V}_\phi(s_2^-, c); \ ...; \ \mathcal{V}_\phi(s_n^-, c)], \tag{5}$$

After this, the Generator assigns a sentence-level ***generative likelihood score*** $\mathcal{G}_{score}$ for each pseudo-statement to reflect how likely the pseudo-statement will be produced under the given context:

$$\mathcal{G}_{score}(c; s_1^-, ..., s_n^-) = [\ell_\theta(s_1^- \mid c); \ \ell_\theta(s_2^- \mid c); \ ...; \ \ell_\theta(s_n^- \mid c)], \tag{6}$$

where $\ell_\theta(s \mid c)$ is the accumulated log-likelihood of the statement $s$ conditioned on the context $c$.

Afterward, each statement with a high Verifier score $\mathcal{V}_\phi(s, c)$ is also expected to receive a high generative score $\ell_\theta(s \mid c)$ to facilitate the Generator's capturing of the Verifier's judgment criterion based on logic consistency. KL-divergence (Kullback & Leibler, 1951) $D_{KL}$ is therefore a appropriate measure for the similarity between the score distribution of $\mathcal{V}_{score}$ and $\mathcal{G}_{score}$. For the purpose of smoothing the gradient to stabilize the GAN training process, we gather both the ground-truth

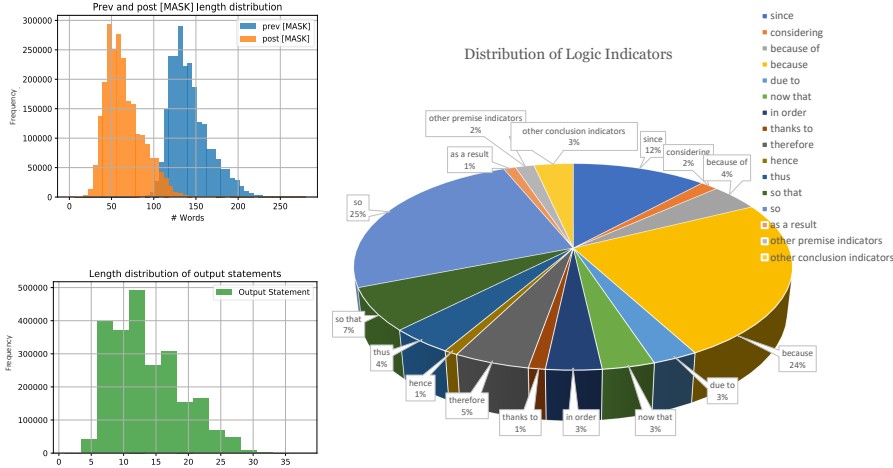

Figure 2: Corpus statistics. Histograms on the left side display length of masked statements (bottom) and prev-and-post statement context (top). The right-side pie chart displays indicators' distribution.

(learned with teacher-forcing loss) and pseudo statements (learned with KL loss) inside the same batch w.r.t. a single input context $c$. In our case, there is exactly one ground-truth statement and $n$ pseudo-statements for each input context $c$. For a batch of $(c; s^+, s_1^-, ..., s_n^-)$, the overall objective of the Generator is defined as (in App. F we show how Eq. 7 commits to the optimization of Eq. 2):

$$\mathcal{L}_{gen}(c; s^+, s_1^-, ..., s_n^-) = \lambda_1\, \mathcal{L}_{tf}(c, s^+) + \lambda_2\, D_{KL}(\mathcal{V}_{\text{score}}(s_1^-, ..., s_n^-)) \,||\, \mathcal{G}_{\text{score}}(s_1^-, ..., s_n^-). \quad (7)$$

## 4 Experiment Setup

### 4.1 Datasets

To test the effectiveness of LogiGAN, we extensively experiment on **12** datasets requiring reasoning via natural language. Specifically, ReClor (Yu et al., 2020), LogiQA (Liu et al., 2021a), Adversarial NLI - ANLI, (Nie et al., 2019), focuses especially on logical reasoning, TellMeWhy (Lal et al., 2021) on abuductive reasoning, HotpotQA (Yang et al., 2018a) on multi-hop reasoning, QuoRef (Dasigi et al., 2019) on reasoing with co-reference resolution, MuTual (Cui et al., 2020), DREAM (Sun et al., 2019)), SAMSum (Gliwa et al., 2019) on reasoning in conversational scenarios, and NarrativeQA (s Kočiský et al., 2018), RACE (Lai et al., 2017), XSum (Narayan et al., 2018) on general verbal reasoning. These datasets make most, if not all, necessary premises for drawing logically consistent conclusions available in their provided context, and require few external premises like commonsense or numerical knowledge. Hence, they fit nicely for testing our hypothesis that LogiGAN brings PLMs logic ability beyond their intrinsic linguistic ability, which could benefit general reasoning processes.

### 4.2 Pre-training Corpus

We apply the corpus construction methodology (§ 2) on the widely used *BookCorpus* (Kobayashi, 2018), which consists of e-books and movies with topics crawled from general domains. Although some corpus featuring debates and arguments (Walker et al., 2012; Abbott et al., 2016; Swanson et al., 2015) appears to be more suitable for our emphasis on logic, we do not elect them due to their high domain specificity in fields such as politics, law, and economics. We discard overly short statements and instances where indicators do not indicate logical reasoning (e.g., "*since* 2010" indicating a time point rather than premises, "*so* happy" indicating degree of the subsequent adjective rather than conclusions). This results in 3.14 million (1.43 and 1.71 million from conclusion and premise indicators, respectively) instances. Corpus statistics are visualized in Fig. 2.

### 4.3 Models

**Baseline Choice.** Since our primary goal of the experiment is to test the effectiveness of LogiGAN and test our hypothesis that logic ability can be further enhanced beyond PLMs' intrinsic linguistic ability, we only compare models pre-trained with LogiGAN against their vanilla versions. After LogiGAN pre-training, we discard the auxiliary Verifier (discussed in Sec. 6) and employ the Generator only to solve all downstream tasks in a purely end-to-end manner. For our main experiments, we initialize Generators from both base and large size pre-trained T5 (Raffel et al., 2020), and Verifier from pre-trained ALBERT-large (Lan et al., 2019). We leave discussions of the rest implementation details and hyper-parameter settings of pre-training and downstream fine-tuning in Appendix D.

**Elastic Search vs. Self Sampling.** As stated earlier in section 3.2, candidate pseudo-statements have two possible sources – they could either be sampled via beam search from the Generator's self-distribution, or could be retrieved from some external resources. We carry out two variant versions of LogiGAN whose Generator is trained purely from self-sampled sentences as pseudo-statements (**LogiGAN** $_{base}$**(ss)**), and from extra pseudo-statements retrieved from corpus by Elastic Search Gormley & Tong (2015) (**LogiGAN** $_{base}$(ss+es)). For the large model, we use LogiGAN $_{large}$ (es+ss) as default. Our database consists of 3.14 million sentences discovered by the corpus construction process, and we keep the top-5 similar retrieved sentences along with self-samples from Generator.

## 5 Experiments

### 5.1 Experimental Results

Table 1: Main results of LogiGAN on 12 downstream tasks (*development sets*).

| Multiple Choice & Classification Datasets | | | | | | | |
|---|---|---|---|---|---|---|---|
| Models / Dataset | ReClor | LogiQA | RACE | DREAM | ANLI | MuTual | Avg. |
| Metrics | Acc | Acc | Acc | Acc | Acc | Acc | |
| Vanilla T5 $_{base}$ | 35.20 | 27.19 | 63.89 | 59.36 | 44.10 | 67.38 | 49.52 |
| LogiGAN $_{base}$ (ss) | 40.20 | 34.72 | 67.13 | 63.38 | 49.50 | 69.41 | 54.06 |
| LogiGAN $_{base}$ (ss+es) | 40.00 | 37.02 | 67.27 | 63.73 | 49.70 | 69.98 | 54.62 |
| Vanilla T5 $_{large}$ | 50.40 | 38.56 | 78.99 | 78.98 | 58.00 | 76.41 | 63.56 |
| LogiGAN $_{large}$ | 54.80 | 40.55 | 80.67 | 81.42 | 63.50 | 77.88 | 66.47 |
| Generation Datasets | | | | | | | |
| Models / Dataset | QuoRef | HotpotQA | NarrativeQA | TellMeWhy | SAMSum | XSum | Avg. |
| Metrics | EM / F$_1$ | EM / F$_1$ | Rouge$_L$ | Rouge$_L$ | Rouge$_L$ | Rouge$_L$ | |
| Vanilla T5 $_{base}$ | 70.76 / 74.58 | 61.11 / 74.86 | 48.11 | 30.03 | 39.32 | 29.14 | 36.65 |
| LogiGAN $_{base}$ (ss) | 75.02 / 78.68 | 62.68 / 76.14 | 49.44 | 31.18 | 39.92 | 30.26 | 37.70 |
| LogiGAN $_{base}$ (ss+es) | 74.94 / 78.40 | 62.80 / 76.18 | 49.46 | 31.15 | 40.21 | 30.27 | 37.77 |
| Vanilla T5 $_{large}$ | 80.06 / 83.25 | 66.11 / 79.80 | 51.09 | 31.42 | 41.40 | 31.58 | 38.87 |
| LogiGAN $_{large}$ | 81.92 / 85.25 | 67.04 / 80.36 | 51.79 | 32.72 | 43.13 | 33.49 | 40.28 |

As presented in Table 1, both base and large size PLMs further pre-trained with LogiGAN surpass their vanilla baselines across both discriminative and generative task formats, through a wide scope of downstream tasks requiring general reasoning abilities. We can make the following observations: Among all observed improvements, those on tasks with particular emphasis on logic (ReClor, LogiQA, and ANLI) are most noticeable. These positive results manifest the effectiveness of LogiGAN in injecting logic ability into PLMs, while testifying to our primary hypothesis that logic ability is fundamental to general reasoning as well. This conclusion answers the two questions in the intro section [7], suggesting that randomized MLM pre-training might fall short in endowing language models with logic ability, and a logic-targeted pre-training approach like LogiGAN may further assist logic learning beyond language acquisition. Furthermore, extra retrieved pseudo-statements (ss+es) bring some additional performance improvement compared with the pure self-sampling (ss) LogiGAN variant, revealing the important role of pseudo-statements' *diversity* in adversarial training.

### 5.2 Ablation Study and Analysis

Observing the apparent performance enhancement, we now aim at pinpointing the truly functional components of LogiGAN through ablation studies and deriving the origins of observed improvements.

---

[7]Is logic ability obtained for free from MLM? Could it be further learned beyond language acquisition?

For fair comparison purposes, we hold all pre-training and downstream settings (including hyper-parameters, implementation designs, and evaluations) unchanged from full LogiGAN. All variations are initialized from $T5_{base}$, and we report performance variance on 7 datasets.

Table 2: Ablation Results on 7 datasets. The last column shows average performance variance, along with relative percentage improvement against vanilla $T5_{base}$ as the baseline.

| Models / Dataset Metrics | ReClor Acc | LogiQA Acc | RACE Acc | DREAM Acc | ANLI Acc | QuoRef EM / $F_1$ | NarrativeQA $Rouge_L$ | — Average |
|---|---|---|---|---|---|---|---|---|
| Vanilla $T5_{base}$ | 35.20 | 27.19 | 63.89 | 59.36 | 44.10 | 70.76 / 74.58 | 48.11 | $49.80_{(+0.0\%)}$ |
| LogiGAN $_{base}$ (ss+es) | 40.00 | 37.02 | 67.27 | 63.73 | 49.70 | 74.94 / 78.40 | 49.46 | $54.59_{(+9.6\%)}$ |
| I. Random Sentence | 36.00 | 30.56 | 61.26 | 58.15 | 45.40 | 70.96 / 74.50 | 48.38 | $50.10_{(+0.6\%)}$ |
| II. MLE Logic Pre-train | 38.80 | 35.02 | 64.55 | 61.71 | 46.00 | 73.61 / 76.96 | 49.30 | $52.71_{(+5.9\%)}$ |
| III. Iterative Multi-task | 37.20 | 34.25 | 64.01 | 62.06 | 46.20 | 71.67 / 75.14 | 49.15 | $52.08_{(+4.6\%)}$ |

**I. Random Masked Sentence Prediction Pre-training.** To explain the observed improvements, our first hypothesis is: Models harvest *extra linguistic ability* from masked *statement* prediction compared with masked *token (or span)* prediction. Quite intuitively, filling entire sentences with complete subject-predicate structures might put additional demands on models to capture more abundant syntactic information beyond the coverage of masked token (or span) prediction. Since LogiGAN involves recovering masked ***sentences***, it is then necessary to determine to what degree, if any, that the observed performance gain is attributable to models' plausible linguistic ability improvement. We therefore carry out a variant pre-training where the prediction objects are *randomly masked sentence*.

Results (shown in Table 2) displays that masked sentence prediction training barely brings improvement against the vanilla baseline. This suggests it is unlikely that masked sentence prediction empowers PLM trained from masked token prediction significantly better linguistic ability, nor likely that the extra pre-training corpus per se significantly raises the performance. Therefore, we reject the first hypothesis and conclude that observed improvements should derive from somewhere else.

**II. MLE-only Logic Pre-training.** Our second hypothesis is that logic-guided masked statement prediction enhances models' intrinsic ability of logical reasoning, thereby lifting the downstream performance. Having addressed the potential impact of learning randomized complete sentence generation, we next aim to check how learning logic-targeted statement generation affects models' behavior. We ablate the entire adversarial training process, and train models to perform maximum likelihood estimation (MLE) with teacher-forcing loss only on masked-out logical statements.

Results 2 of MLE-only logic pre-training reveals quite a notable improvement across almost all datasets against both vanilla baseline and I., suggesting that learning to generate logical statements indeed injects extra abilities into the model. Since results of I. eliminate the possibility that models harvest stronger linguistic abilities from complete sentence prediction, it is safe to partially ascribe the better downstream performance to models' enhanced ability in modeling logical reasoning. This reveals the relative orthogonality between logic ability and models' inherent linguistic ability, suggesting that logic ability could be enhanced through further logic-targeted pre-training.

**III. Iterative Multi-task Pre-training.** Since II. only partially explains the observed improvements, here is our last hypothesis: the adversarial training procedure of LogiGAN explains the unexplained rest part beyond the coverage of II. Here a multi-task pre-training with both generation and verification tasks will be the most natural intermediate setting between the *single-model generation-only setting* of II. and LogiGAN's *dual-model adversarial setting*. However, since the verification task relies on Generator's self-sampled statements, we adopt an iterative self-critic pre-training manner following Nijkamp et al. (2021). Unlike typical multi-tasking training that simultaneously carries different tasks and then sums the losses, our generation and verification tasks happen alternately [8].

Surprisingly, the iterative multi-task pre-training barely brings any positive effects to models' downstream performance compared with II. One possible explanation for this might be that the drastically different mechanisms between the verification and generations task intervene with each other, making the single-model & multi-task setting non-beneficial. Now that we have confirmed that an extra verification task fails to explain the rest improvement, we can accept our final hypothesis and conclude that it is indeed the adversarial mechanism between the Generator and Verifier that truly facilitate learning of logical reasoning, thereby further improving the downstream performance beyond II.

---

[8]Verification is formulated as a generation task – model outputs natural language token "good" and "bad".

# 6 Discussion

**A Psycholinguistic Interpretation of Logic-oriented MLM**   From the first glance, the idea of Logic-oriented MLM seems to be naive and simply. However, we argue that to fully appreciate the value and potential of logic-oriented MLM, going beyond the superficial appearance of masked text and touching down to their underlying psycholinguistic essence is necessary.

It is neither the linguistic patterns of the masked-out text, nor the masking technique to corrupt them that makes logic-oriented MLM (and its potential follow-ups) unique. What truly matters is the distinctive ***cognitive processes*** proceeding in the minds of writers when they put down different pieces of text – language is a window into human minds (Pinker, 2007).

Consider the following examples when a human is filling in the [MASK]'s:

    (1)  "19 + 69 = [MASK]." (Numerical cognition).

    (2)  "Windows is founded by [MASK]." (Declarative memory retrieval).

    (3)  "Socrates is a mortal, so he will eventually [MASK]" (Logical reasoning).

    (4)  "If I feed my dog more than 2 treats per day, it will get [MASK]." (Causal inference).

    (5)  "A crow immediately stands out in swans because it's [MASK]." (Common sense reasoning).

    (6)  "Mike deeply bows to his teacher to show his [MASK]." (Social perception).

Though answers to these [MASK]'s are similar in string length, they nevertheless involve different information pathways and substantially distinctive cognitive processes in writers' minds.

Logic-oriented MLM shines in that it consistently captures exactly one type of such cognitive process, and trains LMs to model humans' logic reasoning mechanism, which is well beyond modeling language per se. On the contrary, Randomized MLM does not capture consistent underlying cognitive processes, which could significantly lower LMs' efficiency of learning advanced intelligence mechanisms beyond language itself. The empirical effectiveness of Logic-oriented MLM provides positive evidence for the practicability of this paradigm, suggesting that LMs might be able to learn various advanced human cognitive processes other than logical reasoning via a similar approach. Combined with LogiGAN's natural analogy of humans' learning-thinking mechanism during logic development, LogiGAN made an encouraging attempt to unify cognitive modeling and language model pre-training.

**Adversarial Training Might Assist Downstream Generation Tasks.**   Although in our experiments, we discard the Verifier and solve downstream tasks with the Generator only, some previous works (Shen et al., 2021; Cobbe et al., 2021) reveal that the Verifier can be used for ranking multiple generation results, thereby effectively enhancing overall downstream accuracy. However, in their paradigm, the information propagates unidirectionally from the Generator to the Verifier, and the Generator cannot directly benefit from the Verifier's discriminative feedback. In contrast, our LogiGAN adversarial training paradigm surmounts the non-differentiable obstacle and could potentially enlighten a new paradigm of both pre-training and downstream fine-tuning.

**Improving Logical Pre-training.**   Our paper demonstrates that PLMs' logic ability can be further enhanced beyond their inherent linguistic ability, and adversarial training may bring extra benefits beyond the learning of logic-targeted masked statement prediction. However, our heuristic-based approach to identifying logical phenomena in a text corpus and the single mask prediction setting can be further improved. Logic recognition methods with higher recall and better unsupervised task designs (e.g., *logical indicator prediction*, or *logic-guided sentence shuffling*) are worthwhile to explore in the further work. Besides, since we are adopting a general domain pre-training corpus (i.e., *BookCorpus*) with bare emphasis on logic, understanding the impacts of extending pre-training to the domain-specific corpus (e.g., law corpus) or others emphasizing logical reasoning is also substantial.

# 7 Related Works

**Generative Adversarial Training in NLP.**   Unlike conventional GAN (Goodfellow et al., 2014; Mirza & Osindero, 2014; Zhu et al., 2017) that generates continuous output such as images, sequential

GAN generates discrete sequences via non-differential searches. This makes feedback from the discriminator not propagatable to the generator. To tackle this challenge, **SeqGAN** (Yu et al., 2017) borrows an idea from reinforcement learning, treating each output token as a single action, and estimates token-wise policy gradient via Monte Carlo search. **RankGAN** (Lin et al., 2017) adopts a similar approach but breaks the binary-classification assumption of discriminator task design, and a ranker provides feedback to the generator. Their generator attempts to generate verisimilar sentences to deceive the ranker into ranking synthetic sentences higher over multiple human-written ones. In our scenario, however, the gold ranking is hard to determine because measuring which statements are more logically consistent w.r.t. context than others is non-trivial, and multi-gold cases are possible. While successfully enabling communication between generator and discriminator, the original designs of SeqGAN, RankGAN, as well as other works such as (Guo et al., 2017; Fedus et al., 2018; Caccia et al., 2018; Rekabdar et al., 2019), generally formulate text generation as a sequential action decision problem, thereby involving heavy sampling for policy gradient estimation, and are sensitive to the length of the target sequence. Since large-scale pre-training (with arbitrary target length) puts a high demand on scalability and computational efficiency, the above approaches are not readily applicable in our scenario. Furthermore, previous work leverages adversarial training to *improve qualities of generated examples*, whereas our focus is on *enhancing models' intrinsic logic ability*.

A recent work, **AR2** (Zhang et al., 2021), leverages adversarial training to improve dense document retrieval. With a retriever-ranker architecture, the learning objective of retriever is to maximize the agreeableness between its own score assignment and that of the ranker for input documents. This is conceptually similar to LogiGAN, as our Generator also aims at reaching consensus with Verifier. However, AR2 does not fall into the sequential GAN paradigm, since it does not involve any sequential text generation, and there is no non-differentiable barrier between the retriever and ranker.

**Pre-training for Reasoning Ability Improvement.**   Previous works have extensively investigated the possibility of injecting specific type of reasoning via pre-training, such as numerical (Geva et al., 2020; Yoran et al., 2021; Pi et al., 2022), commonsense (Zhong et al., 2019; Tamborrino et al., 2020; Staliunaite et al., 2021), formal logic (Wang et al., 2021; Pi et al., 2022), multi-hop (Deng et al., 2021; Zhong et al., 2022), and tabular (Liu et al., 2021b) reasoning. Different from them, LogiGAN focuses on logic reasoning, which plays a fundamental role in general reasoning via natural language.

# 8   Conclusion

In this work, we hypothesize that (i) logic ability plays a key role in a wide scope of tasks requiring general reasoning; and (ii) PLMs' logic ability can be further improved beyond their original linguistic ability. We correspondingly propose LogiGAN, an unsupervised adversarial pre-training framework for logical reasoning enhancement. LogiGAN circumvents the non-differentiable challenge of sequential GAN via a novel Generator-Verifier scoring consensus mechanism, and enables large-scale pre-training with longer target length. Extensive experiments and ablation studies reveal the effectiveness and functional components of LogiGAN, providing evidence to our major hypothesis.

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
