# (Appendix) LogiGAN: Learning Logical Reasoning via Adversarial Pre-training

**Xinyu Pi**[1]*, **Wanjun Zhong**[2]*, **Yan Gao**[3], **Nan Duan**[3], **Jian-Guang Lou**[3]

[1]University of Illinois Urbana-Champaign, Urbana, USA

[2]Sun Yat-Sen University    [3]Microsoft Research Asia

xinyupi2@illinois.edu, zhongwj25@mail2.sysu.edu.cn

{yan.gao, jlou, nanduan}@microsoft.com

## A  Thinking Straight about Reasoning in NLP

The following argument is tentative and is deduced with bounded rationality, and is **for communication purposes only**. We realize and well acknowledge that researchers observing different sets of evidence could hold fundamentally different but reasonable views from ours.

### A.1  Conundrums of Reasoning

Along with the increasing interest in reasoning, multiple reasoning terms are proposed – hybrid reasoning, commonsense reasoning, numerical reasoning, multi-hop reasoning, and unspecified general reasoning, to name a few. However, among all these scattered and distinctive types of reasoning, what is varying? What essence remains constant regardless of the variety of forms? If we were to arrange these reasoning into a hierarchical structure much like a biological taxonomy or to group them into categories, what standard should we follow? How should we put boundaries between each type of reasoning? To the best of our knowledge, few works from the NLP community articulated these queries well. There seems to be a **conceptual conundrum** of reasoning.

Moreover, the ethereal and shapeless nature of reasoning makes it not as visible or concrete as tokens or spans that are readily accessible to the masked language modeling pre-training paradigm. How could we then systematically inject reasoning ability into models via pre-training? What is the nature of this ethereal ability are we truly pursuing? More fundamentally, is pre-training the correct way to add reasoning abilities into language models? Should reasoning abilities be acquired during the pre-training stage, or should it be subsequently tackled by outsourcing symbolic modules (i.e., with neural-symbolic models)? There seems to be a **methodological conundrum** of reasoning.

These questions are non-trivial to answer, and these morals are hard to tell. However, it is quite unlikely that we can solve the reasoning challenge before articulating what the challenge truly is.

### A.2  Reasoning in our Sense

As defined in the introduction section, reasoning (via natural language) is an *inferential process where an unstated statement is drawn based on several presented statements.* Specifically for deductive and inductive reasoning, a conclusion is drawn from provided premises under the guidance of logic. Here is an example of the most trivial cases of such reasoning: given premises "Bob's daughter is called Lily. Lily is now 3 years old.", concluding "Thus, Bob's daughter is 3 years old." requires shallow synthesis between exactly-matched subject and predicate terms.

However, in complex machine reading comprehension tasks, such single-step synthesis can be arbitrarily long-chained (e.g., requires 5 syntheses, and combined with semantic-invariant linguistic

---

*indicates equal contribution. Work done during internship at Microsoft Research Asia.

transformations such as synonym replacements and syntactic transformations. As a result, reasoning over long and rhetorically sophisticated articles becomes non-trivial, demanding highly on both linguistic and logic abilities.

Apart from linguistic transformations, synthesizing among statements usually requires specific synthesis rules of statements. For example, the synthesis rule of degree comparison "Elephants are larger than dogs. Tigers are larger than dogs." → "Elephants are larger than dogs." does not apply to the case "In Pokemon, the water type is strong against the fire type, and the fire type is strong against the grass type." since the conclusion is guided by the same rule "Therefore, the water type is strong against the grass type." fallaciously contradicts reality. Putting into broader domains, mathematics (e.g., arithmetic operators, set operators), formal logic (e.g., quantifiers, logic operators), and most artificial symbolic systems implement their own synthesis rules as standards of correctness. Therefore, while the forms of reasoning remain relatively invariant, the underlying synthesis mechanism could be drastically different from system to system. Among all these symbolic reasoning, the reasoning in NLP focuses primarily on reasoning via natural language – i.e., linguistics as synthesis rules.

### A.3 General Reasoning and Specialized Reasoning

Following the definition of reasoning we make in the previous subsection, we are now able to tentatively categorize all investigations of reasoning in NLP into two families: **Specialized Reasoning and General Reasoning.** We discuss them separately below:

**I. Specialized Reasoning**   can be further divided into sub-categories:

*(a) Reasoning requiring special way of premise extraction*, such as multi-hop reasoning (Yang et al., 2018), tabular reasoning (Zhong et al., 2020), and hybrid reasoning (Chen et al., 2020). The foremost assumption in this scenario is that the input context has already provided all premises necessary to draw targeted conclusions. If we humans are trying to answer a question correctly (i.e., with both correct answers and reasons), we will have to first seek back-and-forth across multiple documents or paragraphs (multi-hop) or rows and columns (tabular) to extract premises for answering this question. Based on these extracted premises, we then follow the logic rules and synthesize a conclusion from these premises to answer the question. Not drastically different from humans, in such complex reasoning scenarios where spurious patterns are mostly unreliable, machines will also have to identify the necessary premises correctly to reach correct conclusions. With all relevant and necessary premises extracted, the rest of the reasoning are reducible to general reasoning described below in II.

*(b) Reasoning requiring external premises*, such as numerical reasoning (Dua et al., 2019), symbolic reasoning (Zhong et al., 2021), domain-specific reasoning (Wang et al., 2022; Gao et al., 2021), and commonsense reasoning (Sap et al., 2020). This family of reasoning requires external knowledge that is harder to be acquired via typical language modeling pre-training. Here we emphasize that knowledge can either be *declarative* or *procedural*, following the theory from psychology (Ten Berge & Van Hezewijk, 1999; Ullman, 2001). For example, the commonsense knowledge "There are 365 days in a year on earth.", or knowledge of historical events can be primarily declarative (i.e., can be articulated with language). Other not readily articulable knowledge, such as swimming, and performing complex arithmetics by hand (e.g. $1969 * 331$) are primarily procedural.

In fact, humans need extra learning to acquire specialized knowledge. For example, we do not acquire knowledge of mathematics or domain expertise for free from language learning. Just as "import PyTorch" is a necessary dependency to implement our neural models in Python, we have to implicitly invoke established mathematics rules to perform calculations, and invoke domain-specific knowledge or laws as premises to make scientific arguments. Machines are generally no different. One special type of premise is commonsense, a large set of knowledge that human effortlessly obtains from multi-modal life experiences. Suppose all necessary premises are readily invoked and stated in natural language, the rest of the reasoning proceeds just as general reasoning, which is discussed below in II.

**II. General Reasoning**   General reasoning covers a broad unspecified form of reasoning that involves recognizing and understanding relevant concepts framed in plain text as premises and then synthesizing them into conclusions. Typically, it assumes stated premises are mostly self-contained (i.e., sufficient for drawing certain conclusions), and require little or no external knowledge to be additionally invoked beyond the given context. Moreover, since premises are presented in a plain

text format, general reasoning does not require special ways of extracting premises from context (e.g., structured data such as tables). The general reasoning is pervasive in tasks emphasizing natural language understanding (NLU), such as machine reading comprehension (Gao et al., 2020) and natural language inference (Nie et al., 2019). General reasoning also serves as an underlying foundation of other specialized reasoning tasks, since most of them can be conditionally reducible to general reasoning, as discussed earlier.

During general reasoning processes, premises can be organized into a logic chain via procedures such as alignment of semantically similar concepts, understanding relations among sentences, and synthesizing sub-conclusions from specific subsets of premises. **Logic** is the systemic set of principles created for providing correctness assurance to conclusions inferred following such logic chains, upon examination of coherence and consistency of these chains [2]. Invocation of logic is therefore necessary for reasoning to be correct (i.e., drawing correct conclusions from correct reasons) – although humans and machines usually carry this out implicitly. Conclusively, logic is the core engine for general reasoning. ***This completes our analysis on logic and reasoning from the introduction section.***

**Some Comments on our Categorization**    Regarding **I-(b)**, we realize it might be less intuitive or even controversial to consider procedural knowledge as "premises". In contrast with crystallized declarative knowledge that resembles more of ***String type variables*** in computer programs, fluid procedural knowledge resembles more of ***functions*** – although functions have well-defined function bodies while procedural knowledge is usually beyond words. Therefore, tasks requiring external declarative knowledge (e.g., commonsense, domain-specific expertise) and procedural knowledge (e.g. symbolic calculations formal logic, arithmetics) might be further divided into two sub-categories.

Apart from the above, our coverage bounds to deductive and inductive reasoning via natural language. Abductive reasoning is the reverse reasoning process where premises are hypothesized to explain or support some stated conclusion, and symbolic reasoning is beyond the coverage of linguistics.

## B    Bridging the Gap between Pseudo and Logical Inconsistency

As mentioned in the footnote of Sec. 3 (The Adversarial Training Framework) from the main text, there is a gap between pseudo-statements that are either self-sampled from the Generator or retrieved and logically inconsistent statements. i.e., there might be logically consistent statements that should have received a positive label when training the Verifier assigned with a negative label since they are pseudo. This could potentially introduce noise signals in the training of the Verifier.

To bridge this gap, we propose a trick that leverages off-the-shelf Natural Language Inference (NLI) models to make judgment on the textual entailment between the pseudo-statement and the ground-truth statement. Our basic intuition here is that: suppose a pseudo-statement implies or is implied by the ground-truth statement, then this pseudo-statement is also expected to be logically consistent w.r.t. the original context, just as the ground-truth one.

For example, in the example of Fig. 1 (LogiGAN Overview), with context, "*All men are mortal, and Socrates is a man. Therefore,* [MASK]*.*", the ground-truth statement "*Socrates is mortal.*" implies the self-sampled pseudo-statement "*Socrates will eventually die.*" from Generator, vice versa. Hence this pseudo-statement should receive a positive label (i.e., $y = 1$) for training Verifier, who learns to discriminate logical consistency of statements w.r.t. input context, instead of merely capturing fake examples. In contrast, the first retrieved pseudo-statement "*a mortal can never be a god.*" does not entail, nor is entailed by, the ground-truth statement. Therefore this pseudo-statement will have a negative label for training Verifier.

Technically, we employ "ynie/albert-xxlarge-v2-snli_mnli_fever_anli_R1_R2_R3-nli" from Huggingface, a well-trained NLI model (denoted as $\mathcal{F}$), to determine the textual entailment relationship between ground-truth and pseudo-statements. The process can be formally expressed as:

$$e(s^+, s^-) = \max(\ \mathcal{F}(s^+, s^-)\ ,\ \mathcal{F}(s^-, s^+)\ ),$$

where $e(s^+, s^-)$ represents the entailment score between the ground-truth statement $s^+$ and a pseudo-statement $s^-$. To determine the final label for training Verifier, we set a hard threshold of $0.50$ – above results in $y = 1$ below turns to $y = 0$. According to our statistical study, this extra NLI mechanism

---

[2]Notice that premises from given context are assumed to be true when solving NLP downstream tasks.

flips around $12\%$ of the pseudo-statements (whose default labels are negative) to $y = 1$. With our human evaluation, the flipped pseudo-statements are indeed logically consistent w.r.t. the original context in most cases.

As an emphasis, the NLI model and the Verifier are making judgment on different things. The NLI does not consider the original context and merely judges the entailment of a statement pair, whereas the Verifier judges one statement's logical consistency w.r.t. the context at a time. The signal from the NLI serves noisy and indirect supervision in Verifier's learning process, and we are leveraging the intrinsic denoising ability of large pre-trained language models. Ablation of this mechanism results in a minor performance drop, but not significant enough to include in our main ablation study, so we are omitting this in the main text.

## C List of Logic Indicators

**Conclusion Indicators (41 in total):** therefore, thereby, wherefore, accordingly, we may conclude, entails that, hence, thus, consequently, we may infer, it must be that, whence, so that, so, it follows that, implies that, as a result, it can be inferred that, suggests that, can conclude, proves that, it can be shown, as a conclusion, conclusively, which implies that, for that reason, as a consequence, on that account, that being said, in conclusion, to that end, for this reason, on account of, because of this, that being so, because of that, ergo, in this way, in this manner, in such a manner, by such means.

**Premise Indicators (20 in total):** since, on account of, considering, because of, because, due to, now that, in order, as indicated by, because, may be inferred from, given that, owing to, by virtue of, owing to, on account of, in view of, for the sake of, thanks to, reason that.

## D Implementation Details

**LogiGAN Details.** We randomly sample 2 million and 0.5 million for source training examples of Generator and Verifier, respectively (i.e., $M = 2$ million, $N = 0.5$ million in Algorithm 1 in the main text). We partition the source Generator training corpus into to two 1 million subsets for warmup and GAN training (i.e., $M_\alpha = M_\beta = 1$ million). In each iteration of GAN training, $10\%$ or 0.05 million and 0.1 million examples are sampled for Verifier and Generator training (i.e., $m = 0.05$ million, $n = 0.1$ million). The warm-up epoch $E$ is set to 5, whereas the max number of GAN iteration is set to 10. For our main experiments, we initialize the Generator from pre-trained "google/t5-v1_1-base" or "google/t5-v1_1-large", and the Verifier as "albert-large-v2" from HuggingFace (Wolf et al., 2020).

We use 8 V100 GPUs for model training, and set the maximum iterations of adversarial training as 15, and set batch size as 8 and 64 for Generator and Verfier training. During Generator training, we put both the only one ground-truth statement, and 5 candidate statements for each instance within the same batch. By default, we adopt learning rate as 5e-5, 1e-5 for the training of Generator and Verifier during adversarial process, respectively.

**Downstream Details.** Our tested downstream datasets mainly belong to two types: generation-based datasets (like extractive QA, abstractive QA, summarization, etc.), and classification datasets (like natural language inference, and multiple-choice QA). We introduce how we process the inputs and outputs for each downstream tasks and the hyper-parameters for fine-tuning.

For the generation-based datasets, we adopt simple hard prompts to write the task input. For example, for the generative QA task, we formulate the input with "*The question is: {question}. The context is: {context}, please give the answer*". For the classification datasets, with the context (optionally question and options) given as the inputs, the target sequence is one of the options, like "entailment" for NLI datasets or one candidate answer for MRC datasets. For example, for the multiple-choice QA task, we prompt the input as "*The question is: {question}. The options are: {options}. The context is: {context}, please select the best option.*", where the output is the specific content of the option. We make the final choice of option by selecting the option with the highest text similarity score calculated by the model output and the context of each option.

During fine-tuning, we don't perform exhaustive parameter searches for each task. We adopt the learning rate as either 1e-4 or 5e-5 for each task, depending on which one will lead to stable

performance. We adopt 8 V100 GPUs for fine-tuning and set batch size as 32 for the base model, and 8 for the large model.

## E  Few-shot Experiments

Table 1: LogiGAN Few-shot Setting Performance.

| Models/Datasets metrics | RACE $Acc$ | DREAM $Acc$ | ReCLor $Acc$ | LogiQA $Acc$ | ANLI $Acc$ | NarrativeQA $Rouge_L$ | $\alpha$NLG $Rouge_L$ | xsum $Rouge_L$ | samsum $Rouge_L$ |
|---|---|---|---|---|---|---|---|---|---|
| Vanilla T5 $_{base}$ | 25.27 | 34.09 | 26.20 | 23.34 | 33.50 | 5.86 | 10.90 | 13.67 | 13.52 |
| Random Sentence | 25.26 | 32.50 | 26.80 | 25.03 | 33.70 | 16.79 | 18.28 | 15.39 | 26.52 |
| LogiGAN-es $_{base}$ | 26.28 | 35.05 | 28.40 | 26.57 | 35.60 | 18.83 | 20.80 | 17.34 | 27.93 |

As a supplementary experiment, we also explore LogiGAN's impact under a data-scarce setting. For each dataset, we randomly select 32 samples and fine-tune models until convergence. To decouple the complete sentence generation and logic learning in the logic-targeted masked statement prediction process, we also add the random sentence pre-training setting into the comparison. This could reveal to what extent the performance variance is explained by the linguistic logic ability improvement separately. Results of provide more evidence for our hypothesis that logic ability and linguistic ability have non-overlapping components.

## F  LogiGAN Training with Policy Gradient

This is to show the overall optimization goal of the adversarial training (Eq. 2 in the main):

$$J^{\mathcal{G}^* \cdot \mathcal{V}^*} = \min_{\theta} \max_{\phi} \mathbb{E}_{s^+ \sim p_{\text{true}}(\cdot|c)} \left[ \log \mathcal{V}_\phi \left( c, s^+ \right) \right] + \mathbb{E}_{s^- \sim p_{\text{neg}}(\cdot|\mathcal{G}_\theta, c, s^+)} \left[ \log \left( 1 - \mathcal{V}_\phi \left( c, s^- \right) \right) \right].$$

is reducible to the optimization problem of KL-divergence in Generator training in (Eq. 7). First of all, we can discard the irrelevant terms of Eq. 2:

$$
\begin{aligned}
J^{\mathcal{G}^*} &= \min_{\theta} \mathbb{E}_{s^- \sim p(\cdot|\mathcal{G}_\theta, c, s^+)} \log \left( 1 - \mathcal{V}_\phi \left( c, s^- \right) \right) \\
&= \max_{\theta} \mathbb{E}_{s^- \sim p(\cdot|\mathcal{G}_\theta, c, s^+)} \log \left( \mathcal{V}_\phi \left( c, s^- \right) \right) \\
&\approx \max_{\theta} \mathbb{E}_{s^- \sim p(\cdot|\mathcal{G}_\theta, c, s^+)} \mathcal{V}_\phi \left( c, s^- \right).
\end{aligned}
\tag{1}
$$

Since the sampling process of $s^-$ is discrete, we cannot directly optimize the $\mathcal{G}^*$ with gradient descent. Following previous works in Sequential GAN, we apply the policy gradient approach.

$$
\begin{aligned}
\nabla_\theta \hat{J}^{G^*} &= \nabla_\theta \mathbb{E}_{s^- \sim p_\theta(\cdot|c)} \mathcal{V}_\phi \left( c, s^- \right) \\
&= \sum_i \nabla_\theta p_\theta \left( s_i^- \mid c \right) \mathcal{V}_\phi \left( c, s_i^- \right) \\
&= \sum_i p_\theta \left( s_i^- \mid c \right) \nabla_\theta \log p_\theta \left( s_i^- \mid c \right) \mathcal{V}_\phi \left( c, s_i^- \right) \\
&= \mathbb{E}_{s^-} \left[ \nabla_\theta \log p_\theta \left( s_i^- \mid c \right) \mathcal{V}_\phi \left( c, s_i^- \right) \right] \\
&\approx \frac{1}{K} \sum_{k=1}^{K} \nabla_\theta \log p_\theta \left( s_i^- \mid c \right) \mathcal{V}_\phi \left( c, s_k^- \right) \\
&= \nabla_\theta \frac{1}{K} \sum_{k=1}^{K} \left[ -\mathcal{V}_\phi \left( c, s_i^- \right) \log \mathcal{V}_\phi \left( c, s_i^- \right) + \log p_\theta \left( s_i^- \mid c \right) \mathcal{V}_\phi \left( c, s_i^- \right) \right] \\
&= \nabla_\theta - D_{\text{KL}}(\mathcal{V}_\phi \left( c, s_k^- \right), p_\theta \left( \cdot \mid c \right))
\end{aligned}
\tag{2}
$$

which is equivalent to minimizing $\mathcal{L}_{gen}$ as decribed in Eq. 7 from the main text.