# OpenReview forum: "LogiGAN: Learning Logical Reasoning via Adversarial Pre-training"
_NeurIPS.cc/2022/Conference — NeurIPS 2022 Accept_

### Official Review · Reviewer_n4Wt · 2022-06-29

**Rating:** 5
**Confidence:** 4
**Soundness:** 3 good
**Presentation:** 4 excellent
**Contribution:** 2 fair

**Summary:**

This paper aims at improving the logical reasoning ability for language models via a pre-training architecture LogiGAN. The adversarial framework adopts a Generator-Verifier style. The Generator learns to recover the masked statements instead of random tokens/spans in traditional pre-training settings. It also learns to score candidate statements to reach a scoring consensus with the Verifier. The verifier, on the other hand, learns to judge the logical consistency between the context and the predicted statement. Additionally, in this way, LogiGAN overcomes the challenge of the non-differentiability of sequential GAN. LogiGAN is verified on 12 datasets designed to test reasoning abilities and surpasses the baseline model.

The key contribution of this paper, as I understand, is to leverage sententious logical relations in pre-training to enhance the reasoning abilities for language models.


**Questions:**

1. How to ensure that the masked-out statements “are declarative and have a complete subject and predicate structures and are capable of being factually true of false”?
2. How to ensure that the Detection Heuristics indeed reflect ‘conclusion’ or ‘premise’? For example, `so` in “so happy” could be just an adverb indicating degrees.
3. What are the statistics of parameters for LogiGAN? This could be important for a fair comparison.


**Limitations:**

The authors provide an insightful `Discussion` section.

**Strengths And Weaknesses:**

**Strengths**:
1. This paper is overall well-written with clear motivations and good structures.
2. The research topic is interesting, i.e., improving general reasoning ability for language models via taking Logic as an entry point.
3. The proposed LogiGAN is technically sound and surpasses the baseline models in a broad range of reasoning datasets with different kinds of task settings.
4. The ablation studies are nice.

**Weaknesses**:
1. The major weakness of this paper is, in my eyes, the superficial modeling of logical relations. Deductive, Inductive, and Abductive reasoning types are only manifestations of logical reasoning, they do not touch down to the essence. Based on this classification, this paper only models the consequential relations in a sentence-level style, which ignores a broad range of other logical reasoning expressions such as negation, disjunction, and conjunction.
2. The so-called automatic identification of logical reasoning phenomena is in fact requires many manual labors. The authors have to pre-define words/expressions that indicate conclusion or premises, which greatly limits the universality of the proposed approach.
3. The authors only report the experimental results on the development set, which made the results less convincing. Also, the performance improvement over generation datasets is less noticeable than in classification datasets. It's hard to know that the model is really behaving as advertised rather than just adding capacity on top of the baselines.
5. In the related work section, especially `Pre-training for Reasoning Ability Improvement`, the authors describe several papers of other relation types that LogiGAN is not compared with. This makes it difficult to place this work in comparison with prior literature and understand the novelty of the proposed framework. Also, the key contribution of this paper could be blurred with improvement over GAN-like architecture rather than pre-training tasks or something else.
6. I do not see any open-source statements or code submissions in the supplementary materials, so I take issues with the reproducibility of this paper.

**Minor Issues**:
1. Line 19: “Learning without thought is labor lost” -> “learning without thinking is labor lost”
2. Caption of table 1: “development setsjudgement” -> “development set judgment”

Overall, I'm borderline with this paper. I think the paper tests three aspects: 1) GAN-style training in language models. 2) predict the masked-out logical statements as the pre-training task. 3) detection heuristics for logical reasoning phenomena. It certainly demonstrates great experimental results compared with vanilla T5 baselines. But none of them **alone** is novel enough. The most important part in my eyes, leveraging logical reasoning phenomena for pre-training, is after all a sententious logical relation extraction, whose effectiveness is already demonstrated in [1].

[1] Huang, Yinya, et al. "DAGN: Discourse-aware graph network for logical reasoning." NAACL 2021.

---

> ### Author Response · Authors · 2022-08-02
> **Response to concerns and questions**
>
> Thanks for recognizing our writing, motivation, and promising results. However, it seems that there are several misunderstandings, which will be clarified in the following. We would really appreciate it if you could re-evaluate our work with our clarifications.
>
> **W1: Logical Relation Modeling**
>
> As presented in Line 86-94, this work focuses on **informal logic** reasoning (deduction, induction, abduction, etc.) defined by Frans H. [1]. The fundamental differences between informal and formal logic can be summarized as:
>
> 1. Formal logic emphasizes the functioning of quantifiers and logic operators, whereas informal logic focuses on the underlying logical mechanism expressed in ordinary natural language.
>
> 2. Compared with formal logic, informal logic is more pervasive in general reasoning tasks such as reading comprehension. Limited number of sentences in text corpus from general domains can be expressed in formal logic expressions, while informal logical phenomena can be identified widely. Moreover, formal logic could also be implicitly entailed in informal logic phenomena. Like the example in Fig. 1, the informal logic statement “all men are mortal” entails the quantifier "all".
>
> 3. The strict format of formal logic expression makes it harder to be accurately identified from the large-scale corpus in an unsupervised manner, whereas logical indicators can provide a firmer way to identify informal logic phenomena.
>
> Since the objective of LogiGAN is to enhance the general logic ability of PLMs through unsupervised pre-training and benefit a wide variety of general reasoning tasks (not only logical reasoning), we believe that modeling informal logic is the most suitable practice.
>
> [1] Frans H. van Eemeren (2009).  "The Study of Argumentation"
>
> **W2：Manual Efforts**
>
> As logic indicators are universal across natural languages, we only need to define them once for each language. Linguists already annotated a lot of indicators in previous studies, so what we need to do now is to expand the list via a thesaurus. Compared with the scale of the pre-training corpus we built in an unsupervised manner, the tiny efforts involved in the whole annotation and verification process are negligible.
>
> **W3: Performance Evaluation**
>
> We would like to respectfully clarify that it is unfair to compare the numerical magnitude of two different metrics. Classification tasks output a single label as the answer (metric: accuracy), while the generation tasks output free-formed text (metric: Rouge-L). Generally, the average absolute performance gain for generation tasks is lower than that for classification tasks.
>
> Under perfectly controlled comparison (hyperparameters, training design, and even random seeds are all kept the same), LMs demonstrate pervasive performance gain on 12 datasets, and an obvious progressions along Vanilla (0.0%) -> Logic-oriented MLM (+5.9%) -> Adversarial (+9.6%) in Table 2. Therefore, even if we didn’t make test-set submissions due to onerous submission workflows, we still have solid evidence that LogiGAN endows LMs with substantial ability improvement that is more than statistical contingency.
>
> **W4: Model Comparison**
>
> The relation between LogiGAN and listed related works is that all of them focus on unsupervised continual pre-training of PLMs for some specific types of reasoning. Different from them, LogiGAN makes the first attempt to improve PLMs’ intrinsic abilities of informal logic -- the foundation of general reasoning (Appendix A). Specialized reasonings, such as numerical and hybrid reasoning, requires models to possess specialized knowledge to perform correctly, which is out of the scope of informal logic and should not be covered in our experiment.
>
> **W5: Open-source Statement**
>
> The open source statement was placed in the footnote on Page 1 in the **original** submitted version.
>
> **Novelty**
>
> To the best of our knowledge, LogiGAN is the first attempt to conduct unsupervised large-scale adversarial pre-training for logical reasoning. And LogiGAN remains fundamental differences from DAGN, which models discourse structures and furnishes LMs with extra information from graph-represented sentential discourse relations. LogiGAN trains LMs to model humans' cognitive process of logical reasoning, which is well beyond the modeling of sentential relations. Logic-oriented MLM has unique novelty and can be generalizable to broader advanced cognitive processes other than logic, such as causality and numerical cognition (**Appendix-G**).
>
> **Q1: Identifying Declarative Statements**
>
> In a linguistic sense, we filter out declarative statements via the ending punctuation, which means that we excluded the exclamative and interrogative sentences ending with “?” and “!”
>
> **Q2: Handling Polysemy of Indicators**
>
> As listed in Appendix C, almost all the indicators are monosemy. Besides, we used POS-tag to solve the special case of “so”, excluding the cases when it is "adverb" with the dependency tag of “advmod”.

---

> > ### Comment · Reviewer_n4Wt · 2022-08-08
> > **Thanks for the clarifications.**
> >
> > Thanks for clarifying most of my concerns.
> >
> > I think I have similar concerns as `Reviewer aP9K`. I'm not fully convinced by the author's statement on the novelty of LogiGAN. Actually leveraging GAN-style training has been well explored in the NLP community. And I feel it inappropriate to advertise the proposed pre-training method as **model humans' underlying cognitive process of logical reasoning**. Instead of emphasizing the technical novelty, I think the authors should focus more on the settings, e.g., "they are the first to design pre-training methods to improve LM's ability in logical reasoning of text".
> >
> > Another concern is the propagated error brought in training as the authors use many external tools and manual efforts. For example,  imperative sentences are also ended with ".". How do the authors alleviate its effect in identifying declarative sentences? It would be better if the author could bring out an analysis of how the propagated errors affect the performance.
> >
> > In summary, I do not intend to gatekeep the paper. Given all three positive reviews from the other three reviewers, I decided to raise my score to 5.

---

> > > ### Author Response · Authors · 2022-08-09
> > > **Follow-up Clarifications on the Cognitive Modeling Implications and Error Propagation**
> > >
> > > Thanks for re-evaluating our work and providing many constructive suggestions. We greatly appreciate them.
> > >
> > > We would agree with you that the novelty of LogiGAN relies heavily on its novel setting of "the first attempt to improve PLM's intrinsic logic ability via unsupervised pre-training", and hope to add some follow-up clarifications:
> > >
> > >  **1. Novelty Concerns**
> > >
> > > We would like to further clarify that the claim **"model humans' underlying cognitive process of logical reasoning"** is not only about the GAN-like pre-training method. Specifically, there are two major components in the main spirit of LogiGAN: **(1) Logic-oriented MLM and (2) The adversarial pre-training framework**.
> > > It is the **predicting masked-out contents of (1)** that conveys humans' underlying cognitive process of logical reasoning. By predicting these masked-out contents based on their surrounding verbal context, LMs are encouraged to produce utterances not only with *syntagmatic relevancy* as in Randomized MLM, but also with *logical consistency* because of the logic-orientation property of the masked-out contents. To meet this extra demand, LMs learn to capture extra information (i.e., patterns of logical inference) beyond superficial linguistic patterns, in virtue of which extra “logic abilities” are acquired. This interpretation of (1) Logic-oriented MLM is supported by the experimental comparison among vanilla (0.0%) -> ablation-I (+0.6%) -> ablation-II (+5.9%).
> > > The design of (2) can be regarded as an analogical borrowing of a learning strategy (i.e., learning with reflective thinking) that is empirically shown by cognitive psychology to be effective in human learning processes. Through (2), a computational realization of this original human learning strategy is transferred to machine learning and is empirically shown to be useful for LMs’ logic ability development. However, **(2) itself** should not be confused with modeling with **cognitive processes of logical reasoning**.
> > >
> > > Lastly, we intend to give more insights into the potential of cognition-oriented MLM. Logic-oriented MLM could be an umbrella project for a line of research focusing on generalized cognition-oriented MLM (e.g., numerical reasoning, common sense, causality, social perception) that enhances models’ intrinsic procedural reasoning abilities in different aspects. **Appendix-G** demonstrates concrete examples and articulates our rationale in-depth.
> > > For example, one can inject "numerical reasoning ability" by predicting "49+16= [MASK]" and "I'm 3 years old, and he is 1 year older than me. He is [MASK] years old."
> > >
> > >
> > > **2. More details on Error Propagation**
> > >
> > > By taking a closer look at the comprehensive indicator list presented in Appendix C to discover that almost all of them are in fact monosemy. This is because we consciously avoided indicators (such as the conjunction “as” as premise indicator) whose senses are overly hard to disambiguate. As a result, there are only two indicators, namely “so” and “since” that must be treated with extra caution among all selected indicators.
> > >
> > > First, POS-tags and dependency tags are quite useful as an initial filter. When leveraged as degree descriptors (e.g., “so happy”), the POS-tag of “so” is **adverb** and dependency tag is “advmod”, whereas the POS-tag of conclusion indicator “so” is **conjunction** (and dependency tag “advcl-mark”) and links two coordinating sentences. The same rule also applies to “since” (e.g., “since 1969”, “since the time that …”), as it displays distinct linguistic behaviors when having different word senses. Empirically, this resort has already handled more than 80% of ambiguities (according to sampling and human evaluations).
> > >
> > > We further add sentence length constraints and syntactic constraints to secure the quality of masked clauses. Overly short (i.e., length < 5) clauses are excluded, and whatever units that lack wholistic subj-predicate (e.g., subj-verb-obj or subj-cop-adj) structures are excluded. Through these heuristics, the only indiscernible corner case is when “since” serves as a marked word of adverbial clause (e.g., “I have been learning Latin *since* I was very young”). However, this minority usage is unlikely to introduce noises significant enough to produce the "error propagation effect", since PLMs possess intrinsic denoising capability that could expectably handle them.  Moreover, since the objective of unsupervised pre-training is to learn general reasoning ability before
> > > fine-grained adaptations to specific downstream tasks, we want to emphasize that **extremely high precision is typically not a necessity in large-scale pre-training, as in sharp contrast with fine-tuning on supervised downstream tasks**.
> > >
> > >
> > >
> > >
> > > To sum up, it would be nice if we could clarify major misunderstandings or confusions regarding the novelty and technical details of LogiGAN, so as to reveal its soundness and potential values. Thanks for the efforts of reviewing and your insightful suggestions again!

---

### Official Review · Reviewer_1vTP · 2022-07-11

**Rating:** 6
**Confidence:** 1
**Soundness:** 2 fair
**Presentation:** 2 fair
**Contribution:** 3 good

**Summary:**

The paper presents a LogiGAN, a generative adversarial approach inducing logic reasoning language models. The core claims made in this paper are: (a) logic helps with reasoning, (b) adversarial pre-training improves performance on NLP tasks requiring reasoning, (c) Generator-Verifier mechanism enables large scale pre-training with arbitrary target length. The claims are supported by experiments over 12 datasets and an ablation study.

**Questions:**

- 330-331: What is the difference between 'qualities of generated examples' and 'intrinsic logic ability'?
- What is 'x' in equation 1? Is it supposed to be 'c'?

**Limitations:**

The authors have not mentioned either. The work does not seem to pose any threat to society beyond those inherent to a large language model.

**Strengths And Weaknesses:**

The paper has an original approach to GAN for language models that demonstrates improvements of results. The quality of the paper is in its evaluation on 12 downstream tasks, although it would be nice to see a comparison with SOTA on the given task for reference.

Strengths:
- The paper is well written and presents consistent improvements over all experiments.
- Novel way of circumventing the non-differentiability issue of SeqGAN

Weaknesses:
- The paper claims that their method allows for large scale pre-training with arbitrary target length. This is not clear to me since (a) the verifier is based on a ALBERT-large and therefore, as far as I understand, is still limited to 512 tokens, (b) I would expect a more detailed explanation as well as empirical results of how much more efficient the LogiGAN is compared to sequential methods to substantiate this claim as a contribution.
- The paper claims that linguistic and logic ability is orthogonal. However, the evidence for this comes from Ablation I, where the model is trained to predict random masked sentences. I do not understand how this demonstrates that linguistic ability does not play a role in improving the model performance. The random selection of masked sentences surely removes logical and linguistic merit equally. Furthermore, in Ablation II, it is showed that the model does considerably improve with MLE pre-training.

Formatting:
- Figure 2 is hard to read in print
- Table 1 has a typo in the title (devleopment setsjudgement).
- paragraph names should end with a period
- 75- GAN posed (GAN put*)

---

> ### Author Response · Authors · 2022-08-02
> **Clarification on the ablation study & Response to other questions**
>
> Thanks for appreciating the writing and novelty of our paper. We will address your questions as follows.
>
> **W1**:
>
> **Clarification of target length**
>
> We made this claim in comparison with the limited acceptable length of SeqGAN.
> We agree that our original wording may be misleading, and we really appreciate the reviewer for pointing that out. In our revision, we changed this claim for clarification.
>
> **Efficiency comparison**
>
> LogiGAN conducts an adversarial process in the large-scale pre-training stage with variant target length and a large training corpus. And due to low efficiency (e.g., SeqGAN can accept limited target output with 5 words), the sequential methods cannot be directly adopted in this scenario. Therefore, it is hard to directly implement the methods in the pre-training setting. It is evident that LogiGAN improves efficiency by avoiding the Monte Carlo search process, whose complexity increased exponentially with target length.
>
> **W2: Orthogonality between linguistic and logical ability**
>
> “Linguistic ability” refers to the capacity to properly model syntactic structure, grammatical rules, semantic implications, and pragmatic context. Since PLMs are generally trained to predict masked-out **tokens** (without holistic syntax, grammar, or sentential semantics), models’linguistic abilities can intuitively be improved from training to predict **randomized** complete sentences.
> However, better linguistic ability does not directly entail better reasoning abilities. For example, under the premise “All men are mortal, and Socrates is a man. Therefore, [MASK]”, subsequent predictions (1) “Socrates will eventually die.” and (2) “Socrates has already died.” are both correct in **linguistic sense**, but (2) is illogical. In short, perfect linguistic correctness does NOT ensure logical correctness.
>
> It is the gap (or “orthogonality”) between linguistics and logic that makes PLMs fall short in downstream tasks emphasizing reasoning procedures. With this rationale in mind, Ablation-I trains models to make MLE predictions on randomly selected complete sentences and aims to equip PLMs with better linguistic abilities **only**. In contrast, Ablation-II trains models to make MLE predictions on sentences with reasoning processes marked by logic indicators (line 95-100), and it is expected to endow models with extra **logic abilities** in addition to linguistic abilities in Ablation-I.
>
> According to the experiment results of Ablation-I, only minor downstream performance improvements occur against the vanilla baseline. Therefore, better linguistic abilities (the Appendix few-shot experiment E could serve as strong support for the linguistic abilities gained) cannot explain the +5.9% improvement of Ablation-II. Combined with the fact that the only difference between Ablation-I & II is whether masked-out sentences involve reasoning processes, one should be confident that models indeed acquire some extra logical reasoning abilities.
>
> **Questions:**
>
> **Q1: The difference between "example quality" and "generative ability"**
>
> In our design, Verifier reward is not used to directly improve the quality of the generated example at the current generation step. Instead, it measures the ability of Generator to synthesize logical consistent statements by reaching a scoring consensus with Verifier. Therefore, it improves the intrinsic logic ability of Generator.
>
> **Q2: Meaning of "x"**
>
> Thanks for pointing out this error. We already modified "x" to "c" in the revised version.
>
> **Typos** - Thanks for pointing out these typos, which we addressed in the revision.

---

### Official Review · Reviewer_P5x5 · 2022-07-12

**Rating:** 7
**Confidence:** 3
**Soundness:** 3 good
**Presentation:** 4 excellent
**Contribution:** 3 good

**Summary:**

This paper proposed LogiGAN. It circumvents the non-differentiable challenge of sequential GAN via a novel Generator-Verifier scoring consensus mechanism, and enables large scale pre-training with arbitrary target length. Extensive experiments and ablation studies reveal the effectiveness and functional components of LogiGAN. Experiments on 12 datasets requiring general reasoning abilities, revealing the fundamental role of logic in broad reasoning, as well as the effectiveness of LogiGAN.

**Questions:**

1. Are the vanilla T5 base model's results from your implementation or they are reported by some other people?

**Strengths And Weaknesses:**

1. The paper is well organized and easy to follow.

2. The motivation of this paper is quite attractive. It firstly hypothesizes that (i) logic ability plays a key role in a wide scope of tasks requiring general reasoning; and (ii) PLMs’ logic ability can be further improved beyond their original linguistic ability. Upon automatic identification of logical reasoning phenomena in massive text corpus via detection heuristics, they train language models to predict the masked-out logical statements.

3. The experiments are solid. One concern here is the numbers in table 1 are kind of close. Is that possible to make a significant test for the improvement? Also it will be better to have a column for avg performance.

---

> ### Author Response · Authors · 2022-08-02
> **Response to questions & Extra insight**
>
> Thank you for appreciating the writing, motivation, and promising results of our paper! We will answer your questions in the following section and discuss about extra insights.
>
> **Performance Improvement**
>
> Different task formats employ different evaluation metrics: Classification tasks outputs a single label (metric: accuracy), and generation tasks outputs a free-formed span or long text (metric: EM or Rouge-L). In general, the absolute improvements on generation tasks are less significant (in terms of numerical magnitude, not to be confused with statistical significance) than classification tasks especially using Rouge-L or Bleu score. Therefore, the performance improvements on generation tasks should be considered as more than merely marginal in this sense. **As suggested, we have added average performance variations for two task formats in our revision.**
>
> In regard to the necessity of an extra statistical significance test, we maintain that the noticeable improvements across perfectly controlled comparison (hyperparameters, training designs, and even random seeds are all kept the same) on 12 datasets and the natural performance progressions along Vanilla (0.0%) -> Logic-oriented MLM (+5.9%) -> Adversarial (+9.6%) in Table 2 already demonstrate that LogiGAN endows models with essential and statistically significance, rather than contingent ability improvements to vanilla PLMs. Therefore, extra cross validations may not result in extra information.
>
> **Vanilla baseline**
>
> Since we use T5 to solve all the tasks (i.e., both classification and generation) and many reasoning tasks have no directly reported results, we implement T5-based models using the same hyper-parameter for a fairer comparison. To ensure reproducibility and encourage research on this line, we will release all our code, models, and data as we stated in the footnote on Page 1.
>
> **Further insights**
>
> We would like to make some extra comments regarding the rationale and implications of logic-oriented MLM from a psycholinguistic perspective.
> Logic-oriented MLM consistently trains LMs to model humans' underlying cognitive process of logical reasoning, which is well beyond the modeling of linguistic patterns as performed in randomized MLM training. This paradigm has great potential in teaching PLMs advanced cognitive processes other than logical reasoning. For example, humans’ causational and numerical cognition might be injected into LMs via similar approaches (e.g., “19 + 69 = [MASK]." for numerical cognition). For more concrete details, *please see our appended* **Appendix-G**.
> LogiGAN also involves the adversarial pre-training mechanism to stimulate the "learning-thinking" process of humans.  Results on Table 2 verify that this stimulation can benefit logic learning. This observation could also deliver a deeper understanding of the analogy between LogiGAN’s adversarial mechanism and humans’ learning-thinking cognitive process.

---

### Official Review · Reviewer_aP9K · 2022-07-23

**Rating:** 5
**Confidence:** 4
**Soundness:** 3 good
**Presentation:** 4 excellent
**Contribution:** 3 good

**Summary:**

This paper proposes an unsupervised adversarial pre-training framework called LogiGAN to improve the ability of logical reasoning in pre-trained models. The authors adopt heuristic rules to detect logic indicators and replace the vanilla masked language modeling with masked logical statement prediction. To explore more logically reasonable statements, the authors devise a GAN-like training process which iteratively trains the generator and verifier. Experimental results on 12 datasets show the superior performance of LogiGAN.

**Questions:**

I have included my questions in the weaknesses part.

**Limitations:**

The authors have not adequately addressed the limitations and potential negative societal impact of their work.

**Strengths And Weaknesses:**

Strengths:

1) The motivation to improve the logical reasoning abilities of pre-trained models is intuitive. The proposed method directly supports the motivation.
2) Experimental results show that LogiGAN can outperform T5 on 12 datasets.
3) This paper is overall well-written and easy to follow.

Weaknesses:
1) The novelty of the proposed method is somewhat limited. In my view, the main contribution of this paper includes the pre-training task of masked logical statement prediction and the adversarial training framework. However, masked logical statement prediction seems like a directly improved version of masked language modeling based on logic indicators. And the adversarial training framework which avoids Monte Carlo search and adopts the sentence-level training loss has also been deeply studied in the community of text GANs, such as [1] and [2]. Thus, I feel that the contribution of this paper is mainly on the applicational side.
2) The sampling of pseudo-statements plays an important role in the training of verifier. I wonder why self-sampling and retrieving similar statements based on BM25 can provide high-quality negative samples to benefit the verifier. From my perspective, the negative samples should be logically inconsistent, which are not intuitively reached by these two methods. The authors say that they use a NLI model to assign labels in the appendix. But this only affects the label assignment without improving the quality of negative samples.
3) The authors mask the task-specific texts (which are related to logical reasoning) in the pre-training task, which may degrade the generalization ability of the pre-trained models to general NLP tasks. Other similar works which enhance the pre-trained models with specific abilities mostly conduct experiments on general benchmarks such as GLUE in addition to task-specific datasets, aiming to show that their method can maintain the generalization ability of pre-trained models. I suggest that the authors should also do this to make readers understand the ability of LogiGAN comprehensively.


[1] Maximum-Likelihood Augmented Discrete Generative Adversarial Networks.

[2] ARAML: A Stable Adversarial Training Framework for Text Generation. EMNLP 2019.

---

> ### Author Response · Authors · 2022-08-02
> **Clarifications on novelty & quality of negative examples & evaluation task**
>
> Thanks for appreciating the intuition, writing, and promising results of our paper. However, there appear to be some misunderstandings concerning our paper, which will be clarified in the following parts. We would appreciate it if you could re-evaluate the paper with our clarification.
>
> **W1: Novelty**
>
> To the best of our knowledge, LogiGAN is the first attempt to conduct large-scale unsupervised adversarial pre-training for injecting logical reasoning ability into LMs, and we conduct the comprehensive experiments on 12 tasks to verify the effectiveness of LogiGAN, providing computational evidence to the relative orthogonality between logic and linguistic.
>
> Moreover, Logic-oriented MLM has its core novelty in its psycholinguistic implications, beyond its technical sophistication. In contrast with randomized MLM where LMs learn to model superficial linguistic patterns, logic-oriented MLM trains LMs to model humans' **underlying cognitive process of logical reasoning**, which is well beyond language per se. This paradigm has potential generalizability to broader advanced cognitive processes other than logical reasoning. For example, humans’ causational and numerical cognition might be injected into LMs via similar approaches (e.g., “19 + 69 = [MASK]." for numerical cognition). For more concrete details, please refer to **Appendix-G**.
>
> Regarding the technical aspect of the adversarial mechanism, we appreciate the reviewer for pointing out closely related works and will add them as methodological references. However, the foremost novelty of LogiGAN derives from its analogical simulation of humans' natural learning-thinking adversary during logical reasoning processes. Combined with generalized orientational MLM, LogiGAN might inspire an interdisciplinary line of research on cognition-aware LM pre-training. Also distinctive from traditional GAN, LogiGAN makes judgement based on logical consistency (rather than fakeness) and correspondingly adopts a new sampling strategy. Finally, it efficiently facilitates pre-training via continuous adversarial process.
>
> **W2: Quality of negative samples**
>
> LogiGAN involves three mechanisms to **unsupervisedly** improve the quality of negative samples:
> 1. Improved Generator enhances the quality of self-sampled negative examples. - The generative ability of the Generator is gradually enhanced during the adversarial training process so that the Verifier could benefit from harder self-sampled negative examples.
> 2. Retrieved examples improve diversity - Retrieval from real corpus provides diverse and natural negative samples that fit the realistic distribution, while the self-sampled examples reflect the distribution of the Generator.
> 3. NLI model re-assigns logical consistency labels to the sampled examples, which better helps the Verifier identify logical consistency.
>
>
> **W3: Task-specific masking harms generalization ability**
>
> As mentioned in the response to W1, LogiGAN learns logical knowledge via orientational generation. Intuitively, both general linguistic ability (through MLM) and logical ability (through logical statement generation) should be improved during this process.
>
> We agree that testing on general tasks is essential for proving the generalization ability. Therefore, we extensively conduct experiments on 12 benchmarks (Sec. 4.1), covering a wider range of reasoning tasks (including logical reasoning & abductive reasoning & general reasoning in conversation, summarization and question answering), in sharp contrast with previous works specialized in logical reasoning and conducted experiments only on logic-intensive tasks (i.e., LogiQA & ReClor). We have also conducted some backward compatibility tests on simple NLU tasks (e.g., STS-B: 89% -> 91.1%; MRPC: 87.1%->89.5%; and SQuAD: 82.9%->84.36%) and observed consistent improvements against baseline. However, as these tasks do not emphasize complex reasoning, we decide not to include them in our paper.
> Thanks, we will consider testing LogiGAN on wider tasks.

---

> > ### Comment · Reviewer_aP9K · 2022-08-09
> > **Response to Author Rebuttal**
> >
> > Thanks for the clarifications from the authors.
> >
> > Weakness #1 (novelty): The authors clarify the novelty of logic-oriented MLM and adversarial mechanism. But I think it's not contradictory to my original review. Since task-oriented MLM and GANs have been deeply studied in the NLP community, the authors successfully adopt these techniques (with task-specific modifications) to achieve better empirical results in the tasks especially with the need of logical reasoning. Thus, the novelty of this paper is mainly on the applicational side.
> >
> > Weakness #2 (negative samples): The rebuttal may not solve my concerns. As the authors say, LogiGAN makes judgement based on logical consistency. Thus, high-quality negative samples should be logically inconsistent, aiming to benefit the training of verifier. I wonder why self-sampling and retrieving similar statements can explicitly do this.
> >
> > Weakness #3 (generalization ability): I recommend the authors to add the results reported in the rebuttal to the paper (at least in the section of appendix in the next version), which help the readers understand the generalization ability of LogiGAN.
> >
> > In summary, this paper seems a borderline case for me with good empirical results but somewhat limited technical novelty. I also read the comments from other reviewers and find that Reviewer n4Wt has the similar opinion. Thus, I will keep my score as 5.

---

### Meta-Review · Area_Chair_x7UG · 2022-08-26

**Recommendation:** Accept
**Confidence:** Less certain

**Metareview:**

This paper presents a pretraining approach for integrating logical reasoning into language models. The authors develop an adversarial training method where they minimize the difference between the verifier scores and generator scores for logically inconsistent statements, thereby making the model less likely to generate sequences the verifier identifies as logically inconsistent. All the reviewers were positive (to varying degrees) on the contributions of this paper, specifically highlighting the clarity of the method, as well as the empirical rigor and results. Weaknesses pointed out by reviewers included the underspecification of different type of logical relations and the weakness of the pseudo-statement sampling procedure, which doesn't seem to explicitly retrieve logically inconsistent statements as negative examples. Despite these questions, the method is well motivated and clear, and the empirical results speak for themselves. This paper represents a solid piece of work and I'm inclined to see it accepted.

**Award:**

No

---

### Decision · Program_Chairs · 2022-09-14

Accept